# *CCNE1* Is a Putative Therapeutic Target for *ARID1A*-Mutated Ovarian Clear Cell Carcinoma

**DOI:** 10.3390/ijms22115869

**Published:** 2021-05-30

**Authors:** Naoki Kawahara, Yuki Yamada, Hiroshi Kobayashi

**Affiliations:** Department of Obstetrics and Gynecology, Nara Medical University, Nara 634-8521, Japan; yuki0528@naramed-u.ac.jp (Y.Y.); hirokoba@naramed-u.ac.jp (H.K.)

**Keywords:** ovarian clear cell carcinoma, Cyclin-E1, *ARID1A* mutation, synthetic lethality

## Abstract

Background: Ovarian clear cell carcinoma (OCCC) is resistant to platinum chemotherapy and is characterized by poor prognosis. Today, the use of poly (ADP-ribose) polymerase (PARP) inhibitor, which is based on synthetic lethality strategy and characterized by cancer selectivity, is widely used for new types of molecular-targeted treatment of relapsed platinum-sensitive ovarian cancer. However, it is less effective against OCCC. Methods: We conducted siRNA screening to identify synthetic lethal candidates for the *ARID1A* mutation; as a result, we identified Cyclin-E1 (*CCNE1*) as a potential target that affects cell viability. To further clarify the effects of *CCNE1*, human OCCC cell lines, namely TOV-21G and KOC7c (*ARID1A* mutant lines), and RMG-I and ES2 (*ARID1A* wild type lines) were transfected with siRNA targeting *CCNE1* or a control vector. Results: Loss of *CCNE1* reduced proliferation of the TOV-21G and KOC7c cells but not of the RMG-I and ES2 cells. Furthermore, in vivo interference of *CCNE1* effectively inhibited tumor cell proliferation in a xenograft mouse model. Conclusion: This study showed for the first time that *CCNE1* is a synthetic lethal target gene to *ARID1A*-mutated OCCC. Targeting this gene may represent a putative, novel, anticancer strategy in OCCC treatment.

## 1. Introduction

Approximately 239,000 new cases of ovarian cancer and 152,000 deaths due to this disease were reported worldwide in 2012 [1]. The frequency of clear cell carcinoma (CCC) is thought to be 5–10% of all epithelial ovarian cancers in Western countries, but it is higher (>20%) in Japan. Ovarian clear cell carcinoma (OCCC) is resistant to platinum chemotherapy and is characterized by poor prognosis. Therefore, a novel strategy to overcome OCCC is required for a more effective outcome.

The transcription factor hepatocyte nuclear factor 1-beta (HNF-1β) is upregulated in endometriosis and OCCC, suggesting that it might be a key molecule in endometriosis-associated CCC [2]. We previously reported that HNF-1β-overexpressing cells survive by persistent Chk1 activation, facilitated by USP28-mediated Claspin stabilization [3]. Therefore, therapy targeting the HNF-1β-USP28-Claspin-Chk1 pathway could be a novel targeted molecular therapy for HNF-1β-overexpressing OCCC. However, pharmacological inhibition of HNF-1β or Chk1 can cause several adverse effects as they show comparable abundance in numerous organs such as the kidneys, liver, pancreas and digestive tract, and there no available inhibitors of USP28 or Claspin. There are some clinical studies about Chk1 inhibitors in solid tumors. UCN-01 (7-hydroxystaurosporine), the first non-selective Chk1 inhibitor introduced in a phase I clinical study [4,5,6,7], combined with anti-cancer agents, resulted in dose reduction because of its toxicities. In a phase II clinical study, this regimen induced limited activity in triple-negative breast cancer [8]. Although Chk1 inhibitors—for example, AZD7762 [9,10], LY2603618 [11,12], and GDC-0575 [13]—also failed to induce anti-cancer effects as significant as those exerted by single agents, they have the potential, as chemosensitizers, to sensitize cancer cells to a variety of DNA-damaging agents. Therefore, it is encouraged to investigate another anti-cancer target to overcome OCCC.

Today, the strategy of synthetic lethality, a novel concept that utilizes the condition of multiple mutated tumor suppressor genes, is used to induce tumor cell death. Notably, targeted therapy using synthetic lethality has fewer side-effects on normal cells because of its selectivity. In clinical application, the PARP inhibitor is put into practice for hereditary breast cancer resulting from a *BRCA1* or *BRCA2* gene mutation and platinum-sensitive ovarian cancer [14,15,16,17]. DNA can be damaged by environmental factors such as ultraviolet light and radiation, and internal factors such as replication stress. It is considered that a cell acquires at most half a million incidents of DNA damage in a day. The major mechanisms to repair DNA damage are base excision repair and nucleotide excision repair for single-strand damage, and non-homologous end-joining and homologous recombination for double-strand damage. Cancer cells with defective *BRCA1* or *BRCA2* tumor suppressor genes cannot perform homologous recombination. Adding the single-strand repair inhibitor (PARP inhibitor) to this environment results in the accumulation of single-strand damage and induces double-strand damage. While such cancer cells subsequently show cell death, the normal cells, which have normal *BRCA1* and *BRCA2* tumor suppressor genes, carry out homologous recombination repair and escape death [18]. Among the specific mutations associated with OCCC, the *ARID1A*, *PIK3CA* or *PTEN* genes are reported to be highly mutated. We chose to focus on the *ARID1A* gene, which is related to chromatin remodeling [19]. *ARID1A* is the most frequently mutated member of the SWItch/sucrose non-fermentable complex (SWI/SNF complex), a nucleosome remodeling complex. The highest mutation incidence (46–57%) is found in OCCC [20,21,22]. Other tumors harboring *ARID1A* mutations are uterine endometrioid carcinoma (47–60%), ovarian endometrioid carcinoma (30%), gastric cancer (29%), colorectal cancer (5–10%), and pancreatic cancer (3–5%) [23,24,25,26]. The *ARID1A*-constituting SWI/SNF complex controls not only transcription but also replication, repair and cell cycle. The loss of *ARID1A* gene function leads to genomic instability and accumulation of mutations because of inappropriate regulation [20]. Previous studies reported some targets showing synthetic lethal effects on *ARID1A* mutated cancer [27,28,29,30,31,32,33,34,35]. Because almost all of them are concerned with chromatin processing, the *ARID1A*-constituting SWI/SNF complex has a variety of roles. To discover other novel synthetic lethal targets acting in a different manner could contribute to treating many OCCC patients. This study aimed to investigate the candidates for synthetic lethality for the *ARID1A* gene mutation in OCCC. Furthermore, among the candidates obtained by this screening, cyclin E1 (*CCNE1*) was further investigated for its efficacy.

## 2. Results

### 2.1. Screening for Candidate Genes Harboring Synthetic Lethal Effect with ARID1A Downregulation in OCCC

We confirmed the effective interference of *ARID1A* to RMG-I (*ARID1A* wild type line) at 96 h by 5 nM of the siRNA (40.8 ± 14.9 vs. 100.0 ± 6.2, *p* = 0.003) (Figure 1). Figure 2 shows the result of the first siRNA screening. The MTT assay was used to identify the candidate genes whose interference significantly reduced (*p* < 0.05) the cell viability of the *ARID1A*-knockdown group over that of the control group. Among the candidates extracted by the first screening, additional confirmation experiments were conducted. Seven candidates were thus extracted as candidate genes. Some studies reported that the copy number gain of *CCNE1* is associated with aggressive or poor prognosis [36,37] and *ARID1A* mutation is correlated with copy number alteration of *CCNE1* [38]. Six *CCNE1*-related genes play a critical role in suppressing p53 activity, regulating TGF-beta-dependent signaling and then directly or indirectly regulating the cell cycle of cancer cells. By the second analysis, knockdown of *CCNE1* resulted in significantly reduced cell proliferation compared to the control group at 5 nM of the siRNA (*p* = 0.006 and *p* = 0.002). As a siRNA smart pool was constructed by four different sequences of *si-CCNE1*, we next assessed which of the sequences was the most effective by MTT assay using TOV-21G cells, and determined it to be 5’-GUAUAUGGCGACACAAGAA-3’.

### 2.2. CCNE1 Has Synthetic Lethal Effect Only in ARID1A-Mutated Cell Lines

In TOV-21G (*ARID1A* mutation type), the *CCNE1*-knockdown group showed significantly reduced proliferation compared to the control group in a time-dependent manner at 48 h, 72 h and 96 h (89.5 ± 3.0 vs. 100.0 ± 2.5, *p* < 0.001; 63.3 ± 3.9 vs. 100.0 ± 5.4, *p* = 0.009; 62.1 ± 1.3 vs. 100.0 ± 6.6, *p* < 0.001, respectively). In KOC7c (*ARID1A* mutation type), the *CCNE1*-knockdown group showed reduced proliferation compared to the control group in a time-dependent manner at 72 h and 96 h (58.9 ± 4.8 vs. 100.0 ± 4.7, *p* < 0.001; 56.6 ± 3.1 vs. 100.0 ± 6.6, *p* < 0.001, respectively).

In contrast, in RMG-I (*ARID1A* wild type), the *CCNE1*-knockdown group did not show a significant reduction in cell proliferation compared to the control group. In ES2 (*ARID1A* wild type), only at 96 h, the *CCNE1*-knockdown group showed reduced proliferation (92.9 ± 2.1 vs. 100.00 ± 4.6, *p* = 0.014) (Figure 3). To confirm the interference of *CCNE1*, we further assessed the relative *CCNE1* mRNA expression levels between the *CCNE1*-knockdown group and control group by RT-PCR. As a result, *si-CCNE1* was found to sufficiently suppress the mRNA levels of *CCNE1* (Appendix A). We next assessed the *CCNE1* protein expression level. In the case of all cell types, the *CCNE1*-knockdown group showed reduced protein expression compared to the control group at 48 h and 72 h (Appendix A).

### 2.3. Interference of CCNE1 Expression Affects Cell Cycle and Apoptosis

We assessed the effect of interference of *CCNE1* expression on the cell cycle and apoptosis in TOV-21G cells. Knockdown of *CCNE1* showed an increasing trend in the proportion of the sub-G1 phase compared with that in the control group at 48 h and 72 h (9.5 ± 4.6 vs. 8.5 ± 5.4, *p* = 0.819; 19.2 ± 11.5 vs. 8.4 ± 2.5, *p* = 0.186, respectively), and a decrease in the S phase (22.1 ± 2.6 vs. 25.8 ± 4.4, *p* = 0.283; 18.8 ± 3.3 vs. 24.0 ± 1.4, *p* = 0.066, respectively) (Figure 4). Moreover, in the apoptosis assay at 48 h after transfection, the proportion of early and late apoptotic cells significantly increased in the *si-CCNE1* group compared to that in the control group (28.2 ± 3.7 vs. 34.6 ± 1.1, *p* = 0.045) (Figure 5).

### 2.4. Knockdown of CCNE1 Inhibits Cell Proliferation of ARID1A Interfered Cell Lines

Interference of *ARID1A* by 10 nM siRNA showed more effective protein interference than 5 nM at both 48 h and 96 h (Figure 6A,B). To confirm that the interference of *CCNE1* showed selective effects on the *ARID1A* deficient status, we interfered ES2 and RMG-I (*ARID1A* wild type ovarian clear cell carcinoma lines) using *si-ARID1A* (5 nM or 10 nM) and si-control, then 48h after the first knockdown, these cells were transfected by 5 nM or 10 nM of *si-CCNE1*. Setting this secondary transfection time as 0h, we measured cell proliferation chronologically by IncuCyte ZOOM™ for a long time: ES2 for 72 h and RMG-I for 108 h. At the endpoint of these assays, under interference of 5 nM of *CCNE1*, ES2 with *si-ARID1A* (5 nM) and RMG-I with *si-ARID1A* (10 nM), they showed significant cell proliferative suppression compared to the si-control group (98.3 ± 1.3 vs. 83.8 ± 8.1, *p* = 0.034; 96.1 ± 3.9 vs. 57.9 ± 5.1, *p* < 0.001, respectively). Moreover, under 10 nM of *si-CCNE1*, ES2 and RMG-I either with 5 nM or 10 nM of *si-ARID1A* showed significant cell proliferative suppression compared to the si-control group (75.0 ± 7.6 and 76.6 ± 2.4 vs. 98.4 ± 1.6, *p* = 0.004 and *p* < 0.001; 53.1 ± 12.2 and 55.7 ± 5.1 vs. 97.3 ± 3.6, *p* = 0.002 and *p* < 0.001, respectively) (Figure 6C–F).

### 2.5. Interference of CCNE1 Inhibits Tumor Growth in Xenograft Mouse Model

To determine whether the interference of *CCNE1* shows a suppressive effect on tumor growth, we conducted an in vivo assay using a xenograft mouse model. To make murine subcutaneous tumors, 4.5 × 10^6^ TOV-21G cells in 200 μL of PBS were injected subcutaneously in the neck of the dorsal midline in five-to-six-week-old athymic nude mice (as si-control, *n* = 5; as *si-CCNE1*, *n* = 8, respectively). After the tumor palpable point, the *si-CCNE1* with atelocollagen and si-control with atelocollagen were administered subcutaneously once per week for two times. Tumor weight determination in the two siRNA-administered groups indicated that the *si-CCNE1* group showed significantly decreased tumor growth than the control group (452.8 ± 274.6 vs. 789.9 ± 129.0, *p* = 0.013) (Figure 7A). To confirm the effective in-vivo interference of *CCNE1*, mRNA were extracted from the tumor and assessed by RT-PCR. There was no differentiation between the si-control and *si-CCNE1* groups (100.0 ± 15.7 vs. 118.5 ± 38.2, *p* = 0.28). Furthermore, to assess whether *CCNE1* interference has a synergistic effect with anti-cancer agents such as cisplatin (60 µg/kg), a combination in vivo assay was conducted. The *si-CCNE1* and cisplatin (60 µg/kg) with an atelocollagen group showed a trend of decreasing tumor weight compared with the si-control and cisplatin (60 µg/kg) group (274.3 ± 136.4 vs. 420.6 ± 216.4, *p* = 0.191) (Figure 7B).

## 3. Discussion

*ARID1A* is mutated in over 50% of OCCCs and 30% of ovarian endometrioid carcinomas [19]. Over 90% of the *ARID1A* mutations observed in ovarian cancer are frame-shift or nonsense mutations that result in loss of *ARID1A* protein expression [20,21,39]. OCCC ranks second as the cause of death from epithelial ovarian cancer [40] and is associated with the worst prognosis amongst the major ovarian cancer subtypes when diagnosed at advanced stages [41,42]. Furthermore, for advanced stage disease, there is currently no effective therapy.

A previous study reported several synthetic lethal targets involved in epigenetic modification, including polycomb repressive complex 2 (PRC2) catalytic subunit EZH2 [27]; HDAC2, a binding partner of the EZH2-containing PRC2 complex [28]; HDAC6, an epigenetic protein that deacetylates numerous substrates [29]; and BRD2, a bromodomain-containing protein 2 [30]. Some clinical trials targeting the above candidates have been conducted [31,32], and some molecules, such as elesclomol and dasatinib, are reported to have a comparatively selective effect on *ARID1A* mutations [33,34,35]. In this study, we selected *CCNE1* among the seven candidate genes and showed that *CCNE1* is a synthetic lethal target gene to the *ARID1A* mutation in OCCC. The other six candidates relate with *CCNE1* and play a critical role in suppressing p53 activity, regulating TGF-beta-dependent signaling and then directly or indirectly regulating the cell cycle of cancer cells. Further analysis of the six *CCNE1* related candidates is being conducted.

*CCNE1* is a member of the cyclin family and forms a complex with and functions as a regulatory subunit of CDK2, whose activity is required for cell cycle G1/S transition. This protein accumulates at the G1-S phase boundary and is degraded as the cell progresses through the S phase. Many tumors show overexpression of this gene, which results in chromosome instability, and thus may contribute to tumorigenesis. Several studies have reported that *CCNE1* gene amplification or protein upregulation is associated with higher tumor grades and with a worse clinical outcome in a variety of cancers [43,44,45]. Considering only OCCC, *CCNE1* overexpression is reported to occur in 23.3% of the cases [37].

In the current study, interference of *CCNE1* had suppressive a cell proliferative effect on not only *ARID1A* wild type cell lines whose protein expression were downregulated, but also *ARID1A* mutant cell lines. And in-vivo interference of *CCNE1* showed a suppressive tumor growth effect on an *ARID1A* mutant TOV-21G xenograft mouse model. These results suggest that interference of *CCNE1* could have a selective effect on *ARID1A* downregulated or mutated tumor cells. The reason why we could not confirm the *CCNE1* knockdown of tumor obtained from a xenograft mouse model is that it could be influenced by the in-vivo environment, via immune system-related T cells or NK cells. We hypothesized that, because *ARID1A* gene is related to chromatin remodeling [19] and upregulated in the G1-S phase, *ARID1A* deficient cells cannot efficiently handle G1 to S migration, and that DNA damage induced by cisplatin can have a synergetic effect. But the chemo-combination in-vivo assay did not show this effect. More experimentation is required using other anti-tumor agents such as taxane, which targets in different manner.

Our study has some limitations. Firstly, we evaluated the effectiveness of *CCNE1* silencing only using four types of OCCC cell lines. To validate these results and identify that these candidates truly have specific effects on *ARID1A* mutation, further studies should be conducted using knock-in or knockout of *ARID1A*. Secondly, a detailed mechanism of *CCNE1* interference and its synthetic effects on *ARID1A* mutation is still unclear. Therefore, other candidates revealed in this screening should be investigated.

## 4. Materials and Methods

### 4.1. Cell Lines

All cells were maintained in humidified incubator at 37 °C with 5% CO_2_. These cells were maintained in Dulbecco’s Modified Eagle’s Medium/Ham’s F-12 with l-Glutamine and Phenol Red containing 10% fetal bovine serum and 100 U/mL penicillin and streptomycin, and used at a sub confluent status. TOV-21G and ES2 cell lines were obtained from the American Type Culture Collection (ATCC, Manassas, VA, USA). KOC7c and RMG-I were kindly given by H. Itamochi (Tottori University School of Medicine, Yonago, Japan). Among these cell lines, TOV-21G and KOC7c are the *ARID1A* mutant cell type, while RMG-I and ES2 are the *ARID1A* wild type. These cells were maintained in D-MEM/Ham’s F-12 with l-Glutamine and Phenol Red containing 10% fetal bovine serum and 100U/mL penicillin and streptomycin, and used at a sub confluent status.

### 4.2. siRNA Library Screening

We carried out siRNA library screening of human cell cycle regulation-related genes, deubiquitinating enzymes and DNA damage response genes (G-003205, G-006005 and G-004705, Dharmacon^TM^, Cambridge, UK). The RMG-I cell line was grown in six-well plate at a concentration of 4.0 × 10^5^ cells per well and *si-ARID1A* (SI03051461, Qiagen, Hilden, Germany) or si-control (D-001210-02, Dharmacon^TM^, Cambridge, UK) was reverse transfected rapidly at 5 nM according to the manufacturer’s recommended protocol. At 48 h after transfection, *ARID1A*-knockdown and control cells were plated in three wells of a 96-well plate, respectively, at a concentration of 5000 cells per well. In each of the three wells of the *ARID1A*- knockdown and control cells, we transfected 5 nM of the respective siRNA for screening. After 48 h, we measured cell viability by MTT assay (Cell Proliferation Kit I, Roche, Salzburg, Austria) according to the recommended protocol. For each 96-well plate, we transfected the si-control as a negative control, and si-PLK1 (M-003290-01, Dharmacon^TM^, Cambridge, UK) as a positive control [46]. Candidates were extracted as follows. Firstly, difference in cell viability between cells transfected with si-control and *si-ARID1A* was considered to be an effect of *ARID1A* downregulation on cells. Secondly, to avoid cell population error between the two cell groups at the point of assay start, given the cell viability of the negative control group showing normal distribution, we corrected the test results based on the difference with the negative control. Next, for further assessment, the most effective sequence was determined by MTT assay using TOV-21G (Appendix A). The four different sequences of *si-CCNE1* were as follows,

si-*CCNE1*-1: 5’-GGAAAUCUAUCCUCCAAAG-3’

si-*CCNE1*-2: 5’-GGAGGUGUGUGAAGUCUAU-3’

si-*CCNE1*-3: 5’-CUAAAUGACUUACAUGAAG-3’

si-*CCNE1*-4: 5’-GUAUAUGGCGACACAAGAA-3’

### 4.3. Western Blotting

The above cell lines (TOV-21G, KOC7c, RMG-I and ES2) were grown in six-well dish (4.0 × 10^5^ cells per well) and *si-CCNE1* and si-control were reverse transfected at 5 or 10 nM according to the manufacturer’s recommended protocol. Then, we extracted protein at 48 and 72 h after transfection. Samples were applied to Mini-PROTEAN® TGX^TM^ Gels 4–15% and transferred by Trans-Blot® Turbo^TM^ Transfer Pack (BIO-RAD, Hercules, CA, USA). The following antibodies were used for western blotting: primary antibodies against *CCNE1* (#20808, Cell Signaling TECHNOLOGY, San Diego, CA, USA; diluted 1:2000), *ARID1A* (#12354, Cell Signaling TECHNOLOGY, San Diego, CA, USA; diluted 1:2000) and ß-actin (#4970, Cell Signaling TECHNOLOGY, San Diego, CA, USA; diluted 1:10,000). Horseradish peroxidase-conjugated secondary antibodies against rabbit (sc-2004, Santa Cruz Biotechnology, Dallas, TX, USA; diluted 1:10,000) were used.

### 4.4. Real Time PCR

RNA extraction was performed at 24 and 48 h after transfection by a TaqMan Gene Expression Cells-to-CT^TM^ Kit (Invitrogen, Carlsbad, CA, USA) according to the manufacturer’s protocol. PCR was performed on a StepOnePlus^TM^ Real-Time PCR System (Applied Biosystems, Foster City, CA, USA) with 4 μL of cDNA, 10 μL of TaqMan Gene Expression Master Mix (4369016, Applied Biosystems, Foster City, CA, USA), 1 μL of *CCNE1* or GAPDH TaqMan Gene Expression Assay (Hs0126536_m1 or Hs99999905_m1, Applied Biosystems, Foster City, CA, USA) and 5 μL of nuclease-free water (B-003000-WB-100, Dharmacon^TM^, Cambridge, UK), and analyzed by the relative quantitative method.

### 4.5. Cell Cycle Analysis

TOV-21G was grown in six-well dishes (2.0 × 10^5^ cells per well) and *si-CCNE1* and si-control were reverse transfected at 5 nM according to the manufacturer’s recommended protocol. Then, the cells were harvested and washed in phosphate-buffered saline (PBS) before fixation in cold 70% ethanol, which was added drop wise to the pellet while vortexing. Cells were fixed for 30 min at 4 °C. Fixed cells were washed twice in PBS and centrifuged at 250× *g* for 5 min. Cells were incubated with 50 μL of a 100 μg/mL stock of RNase and 200 μL propidium iodide (PI) (from 50 μg/mL stock solution). A BD FACSCalibur (BD, San Jose, CA, USA) flow cytometer was used to analyze the cell population for cell cycle changes.

### 4.6. Apoptosis Assay

The cells were seeded into six-well plates at a concentration of 2.0 × 10^5^ cells per well and then treated with 5 nM of *si-CCNE1* and si-control. Treatment with 1 mM staurosporine was used as positive control for 4 h. A negative control was prepared by incubating the cells in the absence of the agent. After the incubation period, the cells were trypsinized and washed in cold PBS. The washed cells were re-centrifuged, the supernatant was discarded and the cells were resuspended in 1× annexin-binding buffer (Applied Biosystems, Foster City, CA, USA). The cell density was determined and then the cells were diluted in 1× annexin-binding buffer to 1.0 × 10^6^ cells/mL. Then, 5 µL Alexa Fluor^®^ 488 annexin V (Applied Biosystems, Foster City, CA, USA) and 1 µL 100 µg/mL PI working solution was added, and the cells were incubated at room temperature for 15 min. After the incubation period, 1× annexin-binding buffer was added, mixed gently and then the samples were kept on ice. As soon as possible, the stained cells were analyzed by flow cytometry and measuring the fluorescence emission at 530 nm (FL1) and 575 nm (FL3).

### 4.7. Time-Lapse Cell Proliferation Assessment

Cell proliferation was studied using the IncuCyte ZOOM™ Live Cell Imaging system (Essen BioScience, Ann Arbor, Mich., USA) as previously described for kinetic monitoring of proliferation and cytotoxicity of cultured cells [47]. IncuCyte image assays quantify how rapidly the proportion of the area covered by cells increases with time as a function of cell proliferation rate [45]. ES2 and RMG-I cells were seeded into six-well plates at a concentration of 2.0 × 10^5^ cells per well, and *si-ARID1A* (5 nM and 10 nM) was transfected. Forty-eight hours later, each of the cells were seeded in a 96 well plate and secondary transfection using *si-CCNE1* (5 nM and 10 nM) was conducted (0 h). They were transferred to the IncuCyte ZOOM™ apparatus and incubation continued over 72 h or 108 h. In this incubation time, IncuCyte captured images every three hours. After defining the area of the cells, all images were chronologically analyzed focusing on confluence (%).

### 4.8. In Vivo Assay

All animal experiments were conducted according to Guidelines for Proper Conduct of Animal Experiments (1 June 2006, Science Council of Japan) and this study was approved by the animal ethics committee of Nara Medical University (no. 12369, 12406, 12441, 12463, 12513, 12574 and 12619). To generate murine subcutaneous tumors, 4.5 × 10^6^ TOV-21G cells in 200 μL of PBS were injected subcutaneously into the neck of the dorsal midline in five-to-six-week-old athymic nude mice (SLC, Hamamatsu, Japan). First, to confirm the effectiveness of the in vivo siRNA method, we conducted a knockdown of cyclophilin B (PPIB) (D-001136-1, Dharmacon^TM^, Cambridge, UK), a housekeeping gene, and assessed the reduced gene expression of PPIB compared to si-STABLE Non-Targeting siRNA group (D-001700-1, Dharmacon^TM^, Cambridge, UK) at 5 µM and 10 µM complexed with atelocollagen (Koken, Tokyo, Japan). Ten days after the injection, based on palpable tumor, we separated the mice into two groups: si-PPIB group (*n* = 3) and control group (*n* = 3). Reagens were injected once per a week for three weeks (the total injection was three times). Seven days after the last injection, the mice were sacrificed. Samples were immediately preserved in Allprotect Tissue Reagent (76405, Qiagen, Hilden, Germany) and RNA was extracted using an RNeasy Mini Kit (74104, Qiagen, Hilden, Germany) according to the manufacturer’s protocol. Based on the real-time PCR result, which revealed that the knockdown efficacies at 5 µM and 10 µM were 66.7% and 80.5%, respectively, compared to the si-control group, we set the siRNA concentration for the in vivo experiments as 5 µM (Appendix A). We used in vivo HPLC individually ordered as si-STABLE, 5’-GUAUAUGGCGACACAAGAAUU-3’ (CTM-494690, Dharmacon^TM^, Cambridge, UK) and si-STABLE Non-Targeting siRNA. For local administration, 200 µL of the siRNA complexed with atelocollagen at a final concentration of 5 µM, prepared according to the manufacturer’s protocol for local use, was injected subcutaneously around the tumor. Ten days after the injection, based on palpable tumor, we separated the mice into two groups: *si-CCNE1* (*n* = 8) and a control (*n* = 5). Reagens were injected twice per a week for two weeks (the total injection was two times). Seven days after the last injection, the mice were sacrificed. Similar to the above experiment, we assessed the combination with cisplatin in addition to injection of *si-CCNE1* or si-control with atelocollagen around the tumor. Cisplatin (Randa^®^ Inj.) (874291, Nippon Kayaku Co. Ltd., Chiyoda-ku, Japan) was intraperitoneally administrated at 60 µg/kg concentration twice a week.

### 4.9. Statistical Analysis

Data are presented as mean ± SD. Analyses were performed by SPSS v. 25.0 (IBM SPSS, Chicago, IL, USA). To assess the difference between the target group and control, Student’s *t*-test was applied. In case of variables that did not present normal distribution, the Mann-Whitney U test was applied. In multiple comparison, a one-way ANOVA was conducted followed by Grams-Howell. Using the two-way ANOVA method, the synergy effect of cisplatin concomitant use was assessed. Two-sided *p* < 0.05 was considered to indicate a statistically significant difference.

## 5. Conclusions

This study showed for the first time that *CCNE1* is a synthetic lethal target gene to *ARID1A*-mutated OCCC. Targeting this gene may represent a putative, novel, anticancer strategy in OCCC.

## Figures and Tables

**Figure 1 ijms-22-05869-f001:**
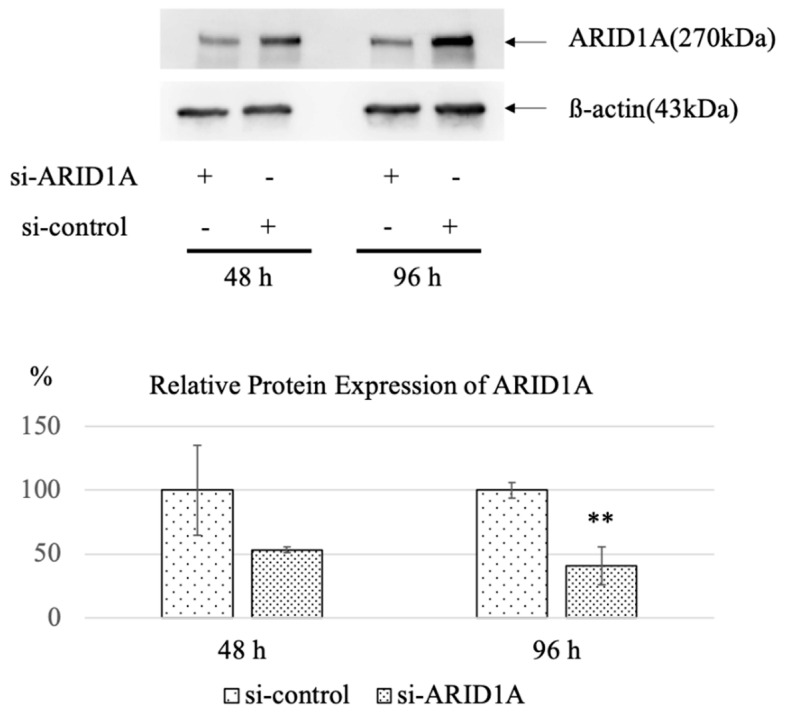
Knockdown efficacy of *ARID1A* interference to RMG-I. Effective interference of *ARID1A* was confirmed at 96 h by 5 nM of the siRNA (*n* = 3 per group). The graphs are described by mean (SD). ** *p* < 0.01 vs. control.

**Figure 2 ijms-22-05869-f002:**
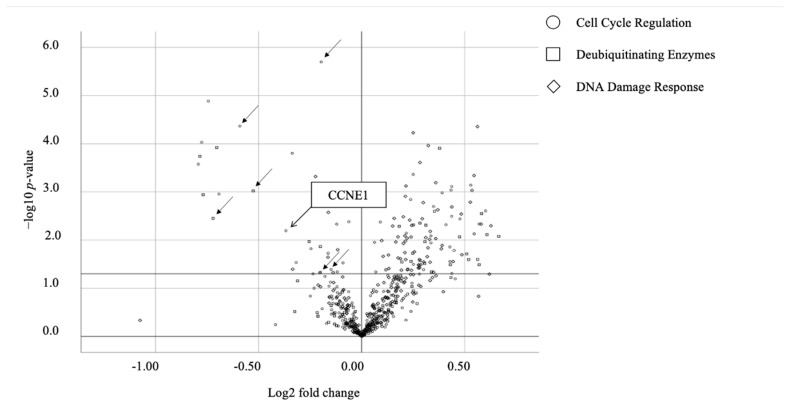
Volcano plot of siRNA library screening using RMG-I. Among 506 genes, seven candidates were extracted; loss of these candidate genes significantly lowered the viability of the *ARID1A*-knockdown cells compared to that of the control cells (*n* = 3 per group). Circle indicates cell cycle regulation-related genes, square indicates deubiquitinating genes and rhombus indicates DNA damage response-related genes. Arrows indicate the seven candidates. The *x*-axis reflects the logarithm of cell survival rate compared to the control.

**Figure 3 ijms-22-05869-f003:**
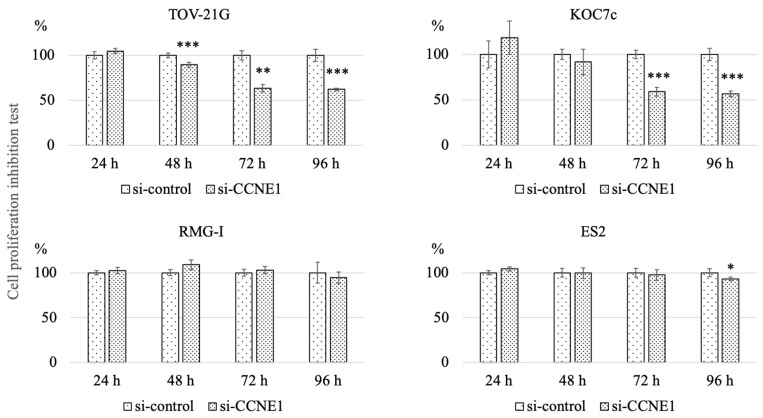
The effect of *CCNE1* interference on cell proliferation. TOV-21G and KOC7c (*ARID1A* mutant cell lines), RMG-I and ES2 (*ARID1A* wild type cell lines) were transfected with *si-CCNE1* or si-control (*n* = 5 per group). In TOV-21G and KOC7c, the *CCNE1* knockdown group showed reduced proliferation compared to the control group in a time-dependent manner, while RMG-I and ES2 cells did not show such reduction upon *CCNE1* interference. The graphs are described by mean (SD). *** *p* < 0.001, ** *p* < 0.01 and * *p* < 0.05 vs. control.

**Figure 4 ijms-22-05869-f004:**
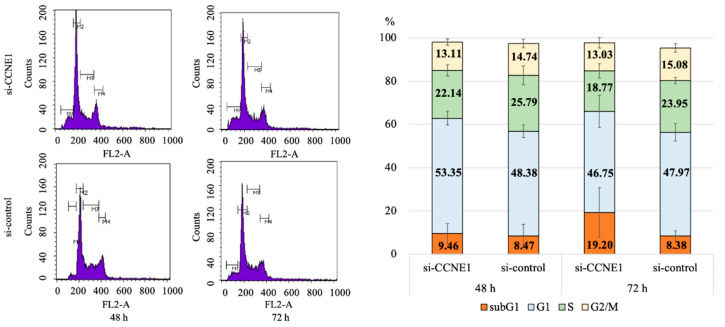
Cell cycle analysis using TOV-21G (*ARID1A* mutant line) under *CCNE1* interference (*n* = 3 per group). *CCNE1* interference resulted in an increasing trend in the proportion of the sub-G1 phase and a decrease in the S phase compared with those in the control group. M1, M2, M3 and M4 refer to the sub G1, G1, S and G2/M phases, respectively. There were three biological replicates, shown by mean (SD).

**Figure 5 ijms-22-05869-f005:**
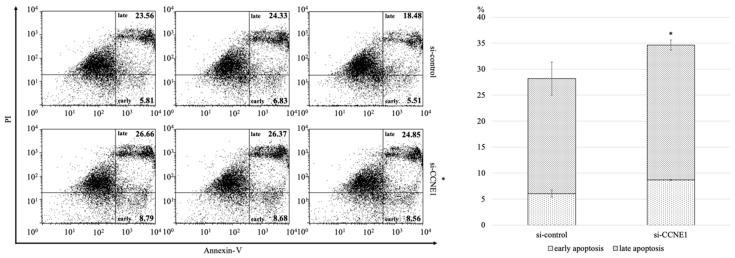
*CCNE1* interference on TOV-21G (*ARID1A* mutant line) increases apoptotic cells. At 48 h after transfection, early and late apoptosis was found to have increased significantly in the *si-CCNE1* group compared to that in the control group (28.2 ± 3.7 vs. 34.6 ± 1.1, *p* = 0.045). There were three biological replicates per group. * *p* < 0.05 vs. control.

**Figure 6 ijms-22-05869-f006:**
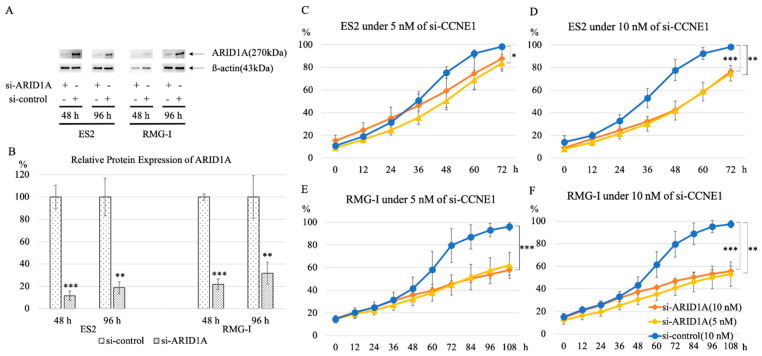
Knockdown of *CCNE1* inhibits cell proliferation of *ARID1A* interfered cell lines. The protein expression levels of *ARID1A*, decreased by 10 nM more in the ES2 and RMG-I cell lines than 5 nM interference (*n* = 3 per group) in macroscopically (**A**) and statistically (**B**). Interference of *ARID1A* in ES2 induced susceptibility to *CCNE1* interference by 5 nM (**C**) or 10 nM (**D**), irrespective of *si-ARID1A* concentration (5 nM or 10 nM) (*n* = 5 per group), and also in RMG-I showed the same result by 5 nM (**E**) or 10 nM (**F**). *** *p* < 0.001 and ** *p* < 0.01 vs. control.

**Figure 7 ijms-22-05869-f007:**
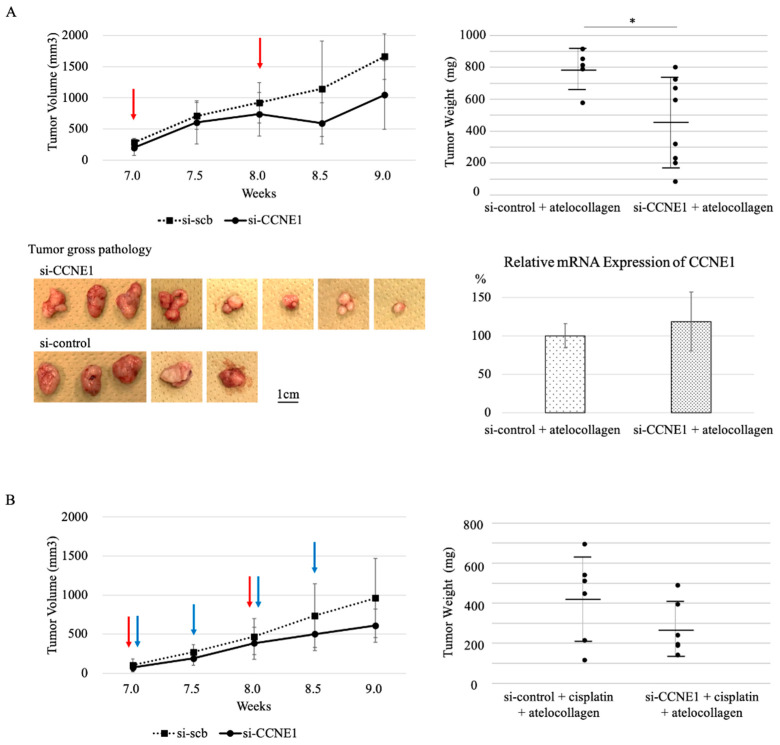
In-vivo efficacy of *CCNE1* interference in TOV-21G xenograft mouse model. *CCNE1* interference has an inhibitory effect on tumor growth. The total number in the si-control group was five, and in *si-CCNE1* was eight. The tumor weight of the *si-CCNE1* group (5 µM) significantly decreased compared with that of the si-control group (**A**). However, the synergy effect of cisplatin (60 µg/kg) (*n* = 6 per group) was not observed (**B**). Red arrow indicates siRNA with atelocollagen injection, and blue arrows indicate cisplatin. * *p* < 0.05 vs. control.

## Data Availability

Not applicable.

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
