# Peer review of "CCNE1* Is a Putative Therapeutic Target for *ARID1A*-Mutated Ovarian Clear Cell Carcinoma"

_ijms, 2021, doi:10.3390/ijms22115869_

Round 1

Reviewer 1 Report

The manuscript describes that downregulation of the cell cycle regulatory gene CCNED1 by siRNA can reduce cell proliferation and induce apoptosis in clear cell ovarian carcinomas with siRNA mediated dowregulation of ARID1 or inactivating ARID1A mutation. The research utilizes two ARID1A mutated and two ARID1 wild-type clear cell ovarian cancer cell lines. The work is largely based on in vitro cell culture experiments, but explores also the concept of CCNED1 downregulation by siRNA in a ARID1A downregulated cell line in a tumor nude mouse transplantation model.

Overall, the authors suggest that targeting CCNDE1 in ARID1A mutated ovarian clear cell carcinomas is a novel approach of achieving synthetic lethality.

The results are interesting and may form a foundation for further studies in exploring this concept of synthetic lethality in clear cell ovarian carcinoma.

I want to express the following comments:

1. Title

The title is overstated. The work is experimental. It is not an in human study. It is based on results with a small number of ovarian cancer cell lines and tumor transplants of one of the cell lines in nude mice.  Therefore, the claim “….a novel therapeutic target for ARID1A…” is too strong. Perhaps, a putative therapeutic target?

2. Results

The results section needs much improvement of the presentation of experiments and results. It must be more carefully and in more detail described what has been done.

a) paragraph 2.1, starting at line 77: introductory sentences should briefly describe the aim and strategy of the siRNA screen, e.g. give a short explanation why you begin with ARID1A downregulation by siRNA although the header of the paragraph states “ARID1A mutation”.

  • Line 79-80: the authors describe “…shows the result of the first si-RNA screening”. How many screens have been performed? Have the results of the different screens be similar?
  • Line 82: please name the seven candidate genes that you have identified in the siRNA screen. Please explain why you choose CCNE1 and not one of the other 6 genes.
  • Lines 85-86: please describe the four different sequences of si-CCNE1 in the Materials and Methods section
  • Figure 1: which cell line has been used. The data in the figure are derived from how many replicates. The graph shows the mean and standard deviation?
  • Figure 2:  the data in the volcano plot have been obtained with which cell line? Are these data from one experiment? How many library screens have been performed? Please give also the name of the other 6 candidate genes in the graph. It would be helpful to explain in the figure legend briefly to what “Log2 fold change” in the x-axis refers.
  • Figure 2 line 97: a writing error - it should be “Arrows” not “Allows”.
  • Figure 3. the graphs show means plus standard deviations? The data are derived from how many replicates? Have the replicates been performed at the same time or independently?
  • Figure 4. Please explain in the legend the meaning of M1-M4. The name of the cell line used is missing. Again - the data are based on how many replicates? The graph shows mean and standard deviation?
  • Figure 5. Please mark early and late apoptotic cell fields. What cell line has been used? How many replicates have been performed? What were the means and standard deviation?
  • Line 137: better insert “siRNA” - Interference of ARID1A by 10nM siRNA….
  • Line 140: please briefly mention what kind of cell lines ES2 and RMG-1 are - ARID1A wild-type ovarian clear cell carcinoma?
  • Line 160: how many mice have been included? One tumor per mouse?
  • Line 166: “…platinum or cisplatin…”. This must be a mistake. Platinum is not an anti-cancer agent - at least not solely platin.
  • Figure 7: photograph of the tumors - please explain these photos in the figure legend. The tumors are derived from how many different mice? Are these all the tumors from the study? Are thes photographs from the si-RNA only treatment or the combination treatment with cisplatin?

3. Materials and Methods

Line 224: “…human cell cycle regulation-related genes….” - this is not the whole story. You employed also libraries to deubiquitinating enzymes and DNA damage response genes.

Line 228: please briefly describe the method of siRNA transfection or at least give a link to the protocoll of the manufacturer. Otherwise, it will be very difficult for readers of the manuscript to find the method description.

LInes 234-235: sorry, I do not understand this sentence. Did you transfect a few wells per every 96-plate with the controls? In this case how many wells with controls?

Lines 235-239: to me this description of the procedure is difficult to understand. It would be helpful to rewrite this paragraph and explain the rationale of siRNA candidate identification in more detail and more clearly.

Line 240: Western blotting - the cell lines are not described. The CCNE1 western blot results are not shown and mentioned in the manuscript. The ARID1A antibody is not mentioned.

What is the siRNA control?

Line 251: Real time PCR - the results of the CCNE1 RT-PCR are not described in the manuscript?

Lines 272-274: this sentence must be wrong? You describe that the cells are fixed in ethanol and then apoptosis is induced. This is impossible. Furthermore, “cell cycle analysis” belongs to the paragraph 4.5.

Line 286: the first sentence should be deleted. The content is repeated in the following sentence.

Lines 295-296: please rephrase and describe in more detail

Lines 307-309: please describe the exact schedule of siRNA injections

Lines 320-322: please describe the exact schedule of siRNA injections

Line 325: Randa is a commercial product name. It would be better to put it in brackets after “cisplatin”

Author Response

Dear reviewer,

Thank you very much for your detailed and kind review.

I felt very appreciated by your comments, and I followed almost all of your suggestions.

It was very suggestive and improved remarkably our manuscript.

Kind regards,

  1. Title

The title is overstated. The work is experimental. It is not an in human study. It is based on results with a small number of ovarian cancer cell lines and tumor transplants of one of the cell lines in nude mice.  Therefore, the claim “….a novel therapeutic target for ARID1A…” is too strong. Perhaps, a putative therapeutic target?

Yes. I would like to change the title.

  1. Results

The results section needs much improvement of the presentation of experiments and results. It must be more carefully and in more detail described what has been done.

  1. a) paragraph 2.1, starting at line 77: introductory sentences should briefly describe the aim and strategy of the siRNA screen, e.g. give a short explanation why you begin with ARID1A downregulation by siRNA although the header of the paragraph states “ARID1A mutation”.

I agree with the comment. I changed the subtitle.

Line 79-80: the authors describe “…shows the result of the first si-RNA screening”. How many screens have been performed? Have the results of the different screens be similar?

Among the candidates extracted by the first screening, additional confirmation experiments were conducted. Knockdown of CCNE1 decreased cell variability twice in the ARID1A interference group. The additional experiment showed a similar results.

Line 82: please name the seven candidate genes that you have identified in the siRNA screen. Please explain why you choose CCNE1 and not one of the other 6 genes.

Yes. I named the seven candidates. And I described the reason why I selected CCNE1.

Lines 85-86: please describe the four different sequences of si-CCNE1 in the Materials and Methods section

Yes. I described.

Figure 1: which cell line has been used. The data in the figure are derived from how many replicates. The graph shows the mean and standard deviation?

The interference efficacy of si-ARID1A to RMG-I was assessed. The replication number is three for each group, and it is described by mean (SD).

Figure 2:  the data in the volcano plot have been obtained with which cell line? Are these data from one experiment? How many library screens have been performed? Please give also the name of the other 6 candidate genes in the graph. It would be helpful to explain in the figure legend briefly to what “Log2 fold change” in the x-axis refers.

The other six candidates showed indication of relation with CCNE1. Six CCNE1-related genes play a critical role in suppressing p53 activity, regulating TGF-beta-dependent signaling, and then directly or indirectly regulate the cell cycle of cancer cells. Currently, we are also researching the functions of these genes, and we will refrain from disclosing specific gene names. We added the comments in the manuscript as possible as we can.

Figure 2 line 97: a writing error - it should be “Arrows” not “Allows”.

I corrected.

Figure 3. the graphs show means plus standard deviations? The data are derived from how many replicates? Have the replicates been performed at the same time or independently?

Because all of the data showed normal distribution, they are shown as mean(SD). The number of biological replicates was five and they were performed at the same time. If required more technical replicates, I can conduct another experiment.

Figure 4. Please explain in the legend the meaning of M1-M4. The name of the cell line used is missing. Again - the data are based on how many replicates? The graph shows mean and standard deviation?

I added the explanation on M1-M4. This assay was TOV-21G (ARID1A mutative). The replication number is three for each group, and it is described by mean (SD). 

Figure 5. Please mark early and late apoptotic cell fields. What cell line has been used? How many replicates have been performed? What were the means and standard deviation?

Yes, I marked. This assay was TOV-21G (ARID1A mutative). The replication number is three for each group, and the mean (SD) was added.

Line 137: better insert “siRNA” - Interference of ARID1A by 10nM siRNA….

I inserted.

Line 140: please briefly mention what kind of cell lines ES2 and RMG-1 are - ARID1A wild-type ovarian clear cell carcinoma?

I inserted a brief explanation.

Line 160: how many mice have been included? One tumor per mouse?

The total number of the si-control group was five, and si-CCNE1 was eight. One tumor grew in one mouse.

Line 166: “…platinum or cisplatin…”. This must be a mistake. Platinum is not an anti-cancer agent - at least not solely platin.

I missed, thus corrected.

Figure 7: photograph of the tumors - please explain these photos in the figure legend. The tumors are derived from how many different mice? Are these all the tumors from the study? Are thes photographs from the si-RNA only treatment or the combination treatment with cisplatin?

The number of mice is described in the figure legend. All these tumors in the photos were delivered from this study. These photos show si-RNA-only treatment.

  1. Materials and Methods

Line 224: “…human cell cycle regulation-related genes….” - this is not the whole story. You employed also libraries to deubiquitinating enzymes and DNA damage response genes.

I missed, thus corrected. Thank you for your indication.

Line 228: please briefly describe the method of siRNA transfection or at least give a link to the protocoll of the manufacturer. Otherwise, it will be very difficult for readers of the manuscript to find the method description.

I gave the link.

LInes 234-235: sorry, I do not understand this sentence. Did you transfect a few wells per every 96-plate with the controls? In this case how many wells with controls?

I prepared six wells each for positive controls and negative controls per every 96-well plate. Six wells were consisted of three wells of si-control cell and three wells of si-ARID1A cell.

Lines 235-239: to me this description of the procedure is difficult to understand. It would be helpful to rewrite this paragraph and explain the rationale of siRNA candidate identification in more detail and more clearly.

Sorry, I modified the sentence more clearly. That was unclear.

Line 240: Western blotting - the cell lines are not described. The CCNE1 western blot results are not shown and mentioned in the manuscript. The ARID1A antibody is not mentioned.

I described.

It seems to be hard to find, the result of CCNE1 western blot results are provided in Supplementary Figure and mentioned in section 2.2., line 119-122.

Sorry, I added the information about the ARID1A antibody.

What is the siRNA control?

We mentioned in 4.2. section, line 238. It is non-targeting negative control RNAi.

Line 251: Real time PCR - the results of the CCNE1 RT-PCR are not described in the manuscript?

It seems to be hard to find, the result of CCNE1 RT-PCR results are provided in Supplementary Figure and mentioned in section 2.2., line 116-119.

Lines 272-274: this sentence must be wrong? You describe that the cells are fixed in ethanol and then apoptosis is induced. This is impossible. Furthermore, “cell cycle analysis” belongs to the paragraph 4.5.

Thank you for your kind pointing out, that was a mistake.

Line 286: the first sentence should be deleted. The content is repeated in the following sentence.

I agree.

Lines 295-296: please rephrase and describe in more detail

I described this process in more detail.

Lines 307-309: please describe the exact schedule of siRNA injections

I described.

Lines 320-322: please describe the exact schedule of siRNA injections

I described.

Line 325: Randa is a commercial product name. It would be better to put it in brackets after “cisplatin”

I changed the sentence according to your suggestion.

Reviewer 2 Report

In this interesting study, Kawahara et al describe siRNA-based screening on ARID1A mutant OCCC cancer type. Considering more than 50% OCCC tumors are bearing ARID1A mutation, this screening for identifying synthetic lethal partner for therapeutic targeting in this deadly ovarian cancer is a very relevant translational research question. The authors identified CCNE1 as a novel synthetic lethal partner in ARID1A mutant OCCC. The authors attempted to validate the results using ARID1A mutant and wild-type OOC cell lines and demonstrate the phenotypic effect in vitro at proliferation, cell cycle, and apoptosis.  Last but not least, they showed the effect of synthetic lethal effect in vivo and tried to combine with platinum-based chemotherapy as a combination treatment strategy. Overall, this is an interesting study for many readers showing the power of siRNA screening to find novel synthetic lethal targets in specific cancer types. However, manuscripts lack some clear descriptions of results and supportive data for the claims. In addition, data representation in the figure can be improved in order to broaden the readership.

Following are the major concerns:

  1. The introduction part of the manuscript can be improved. The second para is more self-citation results with less relevance. A suggestion would be to include some information of findings synthetic lethality results on OCCC and their lack of clinical translation or success after 3rd paragraph (introducing synthetic lethality concept). There should be some motivation to finding a novel synthetic lethal combination in OCCC which is lacking in the current version.
  2. Result section of siRNA screening results (line 77-89, and figure 1 and 2):
    1. lacks information on type of cell line used. Based on material and method, screening was done on ARID1A-wild type RMG-1 cell line in combination of first ARID1A siRNA followed by siRNA screening. This should be clear in the result part and figure legends.
    2. The basis for the selection for CCNE1 among the top 7 genes was lacking.
    3. No information of the top selected 7 genes. I understand the reason for this anonymity; however, it would be important to mention in result section or in the discussion if any of the hit includes previously identified ARID1A synthetic lethal partner to increase the confidence of this single cell line-based siRNA screening.
    4. Unlike screening cell line RMG-1, the selection of specific siRNA from the four siRNAs smart pool was done on Mutant cell line TOV-21G, however, the related data is missing. Kindly provide that at least in supplementary data.
  3. Figure 6: missing data on both concentrations of siARID1A (only 1 concentration was shown for WB and bar plot).
  4. In vivo data, figure 7:
    1. These data lack important siRNA in vivo effect on the expression data of CCNE1 in tumors (either rt-PCR and/or WB)
    2. Figure 7A: need more time points, tumors were harvested at 7 days after two weeks siRNA treatment, there more time points (at least >3 weeks) on tumor measurement should be included (usually tumor size are measured more than once per week in such in vivo
    3. Figure 7 A: 7 mice were used in siCCNE1 group, however, tumor photographs included 8 tumors, this needs to be corrected.
    4. Figure 7B, missing tumor growth data for chemo-combo treatment
    5. Cisplatin dose used in this in-vivo experiment is very low, any rationale to select this dose?
    6. The material method mentioned that pre in-vivo screening for siRNA concentration was done with siPPIB and siSTABLE using rt-PCR. This data should also be included in the supplementary.

  1. In the discussion:
    1. Conclusions mentioned in lines 198-203, about the independence of CCNE1 effect irrespective of ARID1A silencing, lack the data on both siARID1A concentration-dependent ARID1A silencing effect on both wild-type cell lines. Even though if there are differences in ARID1A downregulation at different siRNA concentrations, making this strong conclusion based on transient silencing of ARID1A in WT cell lines and extrapolate it to ARID1A mutant tumors would be an overinterpretation of current data. This needs some textual changes for proper fitting in the discussion
    2. Lacks discussion ARID1A and CCNE1 silencing or mutation effect on proliferation or tumor growth in OCCC or other tumor types
    3. Discussion on in vivo data is lacking and the possibility of chemo-combination based on other published synthetic lethality studies.

Other minor remarks:

  1. In all the figures, missing information on weather data is showing SD or SEM, biological or technical replicate number, statistics info including significance symbol denotation.
  2. In most bar plots, it is preferred to show first control siRNA followed by targeted one for smooth reading.
  3. In figure 4, change the shading of cell cycle bar plots, preferred to add different colors instead of shading for easy distinction of different phases, also missing legend info statistical significance.
  4. In figure 5, missing info from legend if presented three replicates are three different biological replicates or not. The addition of a bar plot with statistics would be easy to read besides flow cytometry plots.
  5. Figure 6: missing information in the legend on the concentration of siARID1A (5nM or 10nM?)
  6. Improved the resolution of figure 6 WB and bar plots and proliferation figure legend keys.
  7. Figure 7 legends lack cell line name and mouse model

Author Response

Dear reviewer,

Thank you very much for your detailed and kind review.

I felt very appreciated by your comments, and I followed almost all of your suggestions.

It was very suggestive and improved remarkably our manuscript.

Kind regards,

Following are the major concerns:

The introduction part of the manuscript can be improved. The second para is more self-citation results with less relevance. A suggestion would be to include some information of findings synthetic lethality results on OCCC and their lack of clinical translation or success after 3rd paragraph (introducing synthetic lethality concept). There should be some motivation to finding a novel synthetic lethal combination in OCCC which is lacking in the current version.

I commented about the existing targets, and our motivation to find novel synthetic lethal targets.

Result section of siRNA screening results (line 77-89, and figure 1 and 2):

lacks information on type of cell line used. Based on material and method, screening was done on ARID1A-wild type RMG-1 cell line in combination of first ARID1A siRNA followed by siRNA screening. This should be clear in the result part and figure legends.

I modified this section.

The basis for the selection for CCNE1 among the top 7 genes was lacking.

No information of the top selected 7 genes. I understand the reason for this anonymity; however, it would be important to mention in result section or in the discussion if any of the hit includes previously identified ARID1A synthetic lethal partner to increase the confidence of this single cell line-based siRNA screening.

The other six candidates showed indication of relation with CCNE1. Six CCNE1-related genes play a critical role in suppressing p53 activity, regulating TGF-beta-dependent signaling, and then directly or indirectly regulate the cell cycle of cancer cells. Currently, we are also researching the functions of these genes, and we will refrain from disclosing specific gene names. We added the comments in the manuscript as possible as we can.

Unlike screening cell line RMG-1, the selection of specific siRNA from the four siRNAs smart pool was done on Mutant cell line TOV-21G, however, the related data is missing. Kindly provide that at least in supplementary data.

Yes. I provided the data in Supplementary Figure.

Figure 6: missing data on both concentrations of siARID1A (only 1 concentration was shown for WB and bar plot).

I added si-RNA concentrations to the figure legend.

In vivo data, figure 7:

These data lack important siRNA in vivo effect on the expression data of CCNE1 in tumors (either rt-PCR and/or WB)

Yes. I added.

Figure 7A: need more time points, tumors were harvested at 7 days after two weeks siRNA treatment, there more time points (at least >3 weeks) on tumor measurement should be included (usually tumor size are measured more than once per week in such in vivo

We measured tumor size twice a week, so changed the figure to more detailed.

Figure 7 A: 7 mice were used in siCCNE1 group, however, tumor photographs included 8 tumors, this needs to be corrected.

Sorry, I missed it. Eight mice were used in the si-CCNE1 group.

Figure 7B, missing tumor growth data for chemo-combo treatment

I added it into Figure 7B.

Cisplatin dose used in this in-vivo experiment is very low, any rationale to select this dose

Because a combination study like ours does not exist, we tried the cisplatin dose as 100µg/kg at first. But this does remarkably decreased mice weight, and sacrifice was deeply concerned. The dose of 25µg/kg was a too small amount in another study (not the current study). From above, we selected this dose.

The material method mentioned that pre in-vivo screening for siRNA concentration was done with siPPIB and siSTABLE using rt-PCR. This data should also be included in the supplementary.

I added the data in the supplementary figure.

In the discussion:

Conclusions mentioned in lines 198-203, about the independence of CCNE1 effect irrespective of ARID1A silencing, lack the data on both siARID1A concentration-dependent ARID1A silencing effect on both wild-type cell lines. Even though if there are differences in ARID1A downregulation at different siRNA concentrations, making this strong conclusion based on transient silencing of ARID1A in WT cell lines and extrapolate it to ARID1A mutant tumors would be an overinterpretation of current data. This needs some textual changes for proper fitting in the discussion

As you suggested, it was relatively strong, and too little evidence I have.

I changed the discussion.

Lacks discussion ARID1A and CCNE1 silencing or mutation effect on proliferation or tumor growth in OCCC or other tumor types

I mentioned in the 4th paragraph.

Discussion on in vivo data is lacking and the possibility of chemo-combination based on other published synthetic lethality studies.

I added a discussion about the in-vivo experiment in the 4th paragraph.

Other minor remarks:

In all the figures, missing information on weather data is showing SD or SEM, biological or technical replicate number, statistics info including significance symbol denotation.

I added.

In most bar plots, it is preferred to show first control siRNA followed by targeted one for smooth reading.

I changed.

In figure 4, change the shading of cell cycle bar plots, preferred to add different colors instead of shading for easy distinction of different phases, also missing legend info statistical significance.

I changed.

In figure 5, missing info from legend if presented three replicates are three different biological replicates or not. The addition of a bar plot with statistics would be easy to read besides flow cytometry plots.

I added detailed information to the legend, and bar plot.

Figure 6: missing information in the legend on the concentration of siARID1A (5nM or 10nM?)

Improved the resolution of figure 6 WB and bar plots and proliferation figure legend keys.

I added. And I modified these figure legend keys. I would consult with the editor about this resolution.

Figure 7 legends lack cell line name and mouse model

I missedit,  and corrected the legend.

Round 2

Reviewer 1 Report

Dear authors,

thank you for revising the manuscript.

Minor comments:

a) the expression "mutative" in the revised manuscript should be substituted by "mutated"

b) Introduction Lines 72-73: please give the references for the mentioned work ("Previous studies reported some targets.......")

c) The seven candidate genes are still not named. However, I understand that the authors do not want to disclose them. Nevertheless, please add that you plan to further characterise the other six genes in further experiments and that for now you concentrated on CCNE1 for the mentioned reasons.

Author Response

Dear reviewer,

Thank you very much for your quick response.

I felt very appreciated by your comments, and I followed all of your suggestions. 

Kind regards,

  1. a) the expression "mutative" in the revised manuscript should be substituted by "mutated"

I changed the term throughout the manuscript.

  1. b) Introduction Lines 72-73: please give the references for the mentioned work ("Previous studies reported some targets.......")

I added the references into the sentence.

  1. c) The seven candidate genes are still not named. However, I understand that the authors do not want to disclose them. Nevertheless, please add that you plan to further characterise the other six genes in further experiments and that for now you concentrated on CCNE1 for the mentioned reasons.

I agree. So I mentioned we concentrated on CCNE1 for this study and briefly referred to the other six candidates and further analysis about them.

Reviewer 2 Report

In the revised manuscript the authors adequately addressed all the issues and concerns raised in the previous version, particularly for the missing additional data and insufficient information for some result parts. However only two minor points are still there:

1) for introduction para 2, include some other authors' work as well in OCCC field instead of self-citation work.

2) Except, in Fig. 7 legend description, it is mentioned that 6 mice were used in siCOntrol group but only 5 tumor images are there and tumor volume data is presented. Need a correction there.

Overall, the authors addressed all my concerns.

Author Response

Dear reviewer,

Thank you very much for your quick response.

I felt very appreciated by your comments, and I followed all of your suggestions.

Kind regards,

1) for introduction para 2, include some other authors' work as well in OCCC field instead of self-citation work.

I agree. Because this study started based on the report of F. Ito et al., who belong to our section, please allow us to cite the study. Thus, we mentioned the limitations the previous study had, and to improve the limitations we were encouraged to start the current study.

2) Except, in Fig. 7 legend description, it is mentioned that 6 mice were used in siCOntrol group but only 5 tumor images are there and tumor volume data is presented. Need a correction there.

It was my mistake. Thank you for your point out.